

# Quantifying the Direct Radiative Effect of Absorbing Aerosols for Numerical Weather Prediction: A case study

Mayra I. Oyola[1], James R. Campbell[2], Peng Xian[2], Anthony Bucholtz[2], Richard A. Ferrare[3], Sharon P. Burton[3], Olga Kalashnikova[4], Benjamin C. Ruston[2], Simone Lolli[5]

[1] American Society for Engineering Excellence, Monterey, CA, 93943, USA.
[2] US Naval Research Laboratory, Monterey, CA, 93943, USA.
[3] NASA Langley Research Center, Langley, VA, 23365, USA.
[4] NASA Jet Propulsion Laboratory, Pasadena, CA, 91109, USA.
[5] CNR-IMAA, Istituto di Metodologie per l'Analisi Ambientale, Tito Scalo (PZ), Italy.

*Correspondence to*: Mayra I. Oyola (mayra.oyola.ctr@nrlmry.navy.mil)

**Abstract:** We conceptualize aerosol radiative transfer processes arising from the hypothetical coupling of a global aerosol transport model and global numerical weather prediction model by applying the U.S. Naval Research Laboratory Navy Aerosol Analysis and Prediction System (NAAPS) and the Navy Global Environmental Model (NAVGEM) meteorological and surface reflectance fields. A unique experimental design during the 2013 NASA *Studies of Emissions and Atmospheric Composition, Clouds and Climate Coupling by Regional Surveys* (SEAC⁴RS) field mission, allows for collocated airborne sampling by the Langley's *High Spectral Resolution Lidar* (HSRL), the *Airborne Multi-angle Spectro Polarimetric Imager* (AirMSPI), up/down SW and broadband IR radiometers, as well as NASA A-Train support from the Moderate Resolution Imaging Spectroradiometer (MODIS), to attempt direct aerosol forcing closure. The results demonstrate the sensitivity of modeled fields to aerosol radiative fluxes and heating rates, specifically in the SW forcing and heating rates, as induced in this event from transported smoke and regional urban aerosols. Limitations are identified with respect to aerosol attribution, vertical distribution and choice of optical and surface polarimetry properties, which are discussed within the context of their influence on Numerical Weather Prediction output that is particularly important as the community propels forward towards inline aerosol modelling within global forecast systems.

## 1 Introduction

Over the last two decades much progress has been achieved in terms of characterizing aerosol properties, identifying their spatio-temporal extent and determining their role in planetary radiative balance (Ramanathan et al., 2001). As a result of that endeavour, the scientific community has been able to recognize aerosols have a *direct effect* on climate by modifying the planet's radiative budget and redistributing heat in the atmosphere, and an *indirect effect* by modifying cloud development, precipitation and optical properties (IPCC, 2014). Additionally, it is implicit that these effects are reliant on aerosol altitude, and on the reflectance (albedo) of the underlying surface (Lyapustin et al., 2011; Bauer and Menon, 2012; Xu et al., 2017).

Nevertheless, significant uncertainty still remains when it comes to understanding the atmosphere's response to different aerosol physical properties, particularly on day-to-day scales that impact weather (Mulcahy et al., 2014, Toll et al., 2016;





Zhang et al., 2016). Aerosols are now regular components of numerical weather prediction models (NWP), and it has been shown through model sensitivity studies that aerosol radiative coupling effects are non-trivial in influencing resolved weather processes (Carmona et al., 2008, Milton et al., 2008; Mulcahy et al., 2014, Toll et al., 2016). For example, increased

aerosol scattering and absorption of incoming shortwave (SW) and outgoing longwave radiation (OLR) fields modify the atmospheric heating profile and can affect both large-scale and regional circulation patterns (Haywood et al., 2005; Mulcahy et al., 2010). Furthermore, the omission of the scattering and absorption properties, in particular for mineral dust and biomass burning, was identified in case study analysis as the principal cause of significant biases (in the order of 50-56 W m$^{-2}$, over dust source regions) in both model OLR at the top-of-atmosphere (TOA) (Haywood et al., 2005) and surface SW

radiation fields (Milton et al., 2008).

Until recently, the representation of aerosols in global NWP systems at most weather offices was based on a simple aerosol climatology or monthly averages of aerosol concentrations and optical properties (e.g. Tegen et al., 1997), which omit the daily variability of these constituents and thus do not account for changes in concentration, size and vertical distribution.

While models show fundamental improvement when considering aerosols (such as the reflected SW radiative bias at the TOA), temperature biases in the lower troposphere of approximately 0.5 K day$^{-1}$ were documented by Mulchany et al. (2010) due to aerosol climatology being too absorbing. These biases, in turn, translate in spatio-temporal discrepancies in precipitation and temperature forecasts (Carmona et al., 2008; Milton et al., 2008).

Examples of significant improvement found in NWP skill when considering aerosols include forecasts of the African Easterly Jet at the European Centre for Medium Range Forecast (ECMWF) and reduction of temperature and precipitation seasonal mean-biases (e.g. Thompkins et al., 2005, Rodwell and Jung, 2008). In the case of real-time or near real-time prognostic aerosols, Mulcahy et al. (2014) demonstrated an overall improvement in the NWP radiative budget fields by means of improved representation of the direct radiative forcing, while Toll et al. (2015, 2016) demonstrated improvement in

forecast of near-surface fields over extreme aerosol events, such as 2010 fires occurring in Russia.

Despite the potential benefits of proper aerosol characterization in NWP systems, aerosols physics have to date not been fully coupled with the operational weather modelling components for a number of reasons, including: a) inaccurate model initialization due to limited knowledge of the aerosol spatio-temporal distribution, particularly in the vertical (Alpert et al.,

2002; Zhang et al., 2011, 2014); b) the physical/chemical effects of aerosols on the atmospheric energy balance, and in particular their various interactions with clouds, are not well constrained (Ramanathan et al., 2001; Jacobson and Kaufman, 2006), and c) added demand on computational requirements. However, advances in data assimilation schemes for NWP applications, combined with the development of accurate, stand-alone, three-dimensional aerosol models, now allows for circumventing some of these limitations. The capabilities for an "in-line" aerosol model, (one that runs in parallel and

coupled with the NWP model) have been developed and implemented at a few centers, including ECMWF ( Mulcahy et al.,



2014) and NASA GMAO (Randles et al., 2017) and is in the process of being implemented at the U.S. Naval Research Laboratory, Marine Meteorology Division. However, such a venture is not trivial and the implications need to be characterized.

Here, we combine operational prognostic aerosol model profiles and a global weather model analysis into a four-stream radiative transfer model, with the express goal of evaluating how well the aerosol model achieves column radiative closure relative to its depiction of the vertical mass concentration profile. We evaluate data generated during the 2013 **S**tudies of **E**missions and **A**tmospheric **C**omposition, **C**louds and **C**limate **C**oupling by **R**egional **S**urveys (SEAC[4]RS), combined with coincident satellite-derived surface reflectance data. The SEAC[4]RS datasets represent a unique opportunity to attempt this

model evaluation experiment, given the instrumentation flown aboard two collocated aircraft that simultaneously measured the vertical aerosol profile at high resolution and airborne up/down broadband solar and infrared fluxes. Thus, we attempt radiative closure with the in situ instrumentation and use it to evaluate model skill in depicting aerosol radiative properties. More specifically, and within the observational constraints of the limited dataset available from which to attempt this study, we aim to understand the magnitude of aerosol heating rates, (particularly those associated with transported smoke), evaluate

surface polarimetry sensitivity to smoke properties and assess the radiative impact of smoke layers and their potential influence on NWP outputs.

## 2 Data and Methods

SEAC[4]RS was conducted in August and September 2013, focused primarily on the south eastern United States; with the
objective of understanding how summer storms and air pollution from wildfires, cities, and other sources impact climate (Toon et al., 2016). As such, a very comprehensive suite of observations from satellites, aircraft, and ground sites were combined, providing a unique opportunity to characterize the radiative effects of aerosols on the basis of their spectral optical properties. In this study, we apply these measurements for process evaluation relative to prognostic aerosol and NWP model fields. Here we describe the tools employed.


### 2.1 High Spectral Resolution Lidar (HSRL)

During SEAC[4]RS, aerosol vertical information was collected by the NASA Langley Research Center (LaRC) airborne Ozone Differential Absorption Lidar and Aerosol/Cloud High Spectral Resolution Lidar-2 (DIAL/HSRL) (Hair et al., 2008), which was flown on the NASA DC-8 aircraft. The NASA Langley airborne HSRL instrument technique has been described
elsewhere (e.g. Hair et al., 2008; Burton et al., 2012, 2013). In short, the HSRL makes direct measurements of aerosol intensive properties, such as aerosol backscatter coefficient ($\beta$) and depolarization ratio ($\delta$) at 355, 532, and 1064 nm wavelengths, and aerosol extinction coefficient ($\alpha$) at 532 nm wavelengths. Data are sampled at 2 Hz and 15 m vertical resolution, which are then horizontally averaged for 10 s ($\beta$ and $\delta$) and 60 s ($\alpha$). However, the nominal resolution for the backscatter and depolarization (extinction) is 30m (300m) in the vertical, and 2km (12km) horizontally.




Of note for this experiment, aerosol characterization using the HSRL-2 instrument is estimated by employing a semi-supervised method based on labeled samples comprising 0.3% of the existing HSRL measurement database at the time of the algorithm development. The labeled samples are cases where ancillary information (e.g., in-situ measurements, back-trajectory analysis, and visual identification of plumes from the aircraft) has been used to determine the aerosol type.

Observations in the remainder of the dataset are then classified by comparison with the labeled samples using the Mahalanobis distance metric (Burton et al., 2012). The HSRL aerosol classification consists of eight types, described by Burton et al., (2012), based on samples of known type observed in airborne field missions in North America since 2006. These are ice, pure dust, dusty mix, maritime, polluted maritime, urban, smoke, and fresh smoke.

**2.2 NAAPS**

Modeled aerosol profiles are based on the Naval Aerosol Analysis Prediction System (NAAPS, http://www.nrlmry.navy.mil/aerosol/, Lynch et al., 2016). NAAPS was developed at the Naval Research Laboratory in Monterey, USA and is a three-dimensional aerosol and air pollution model, originated from a hemispheric sulfate chemistry model developed by Christensen (1997). Dust, sea salt and biomass-burning smoke have been added to the original model,

and are documented in Westphal et al. (2009), Witek et al. (2007) and Reid et al. (2009), respectively. NAAPS runs for this study were conducted in "offline" mode, utilizing meteorological analysis and forecast fields from the 0.3 degree NAvy's Global Environmental Model (Hogan et al., 2014). We apply 550 nm aerosol optical profile information from NAAPS, including $\alpha$ (extinction and aerosol optical depth (AOD), for our radiative transfer simulations.

Currently, NAAPS produces 6-day forecasts of $SO_2$ (gas), anthropogenic and biogenic fine (ABF, combined sulfate and organic aerosols), dust, biomass burning smoke and sea salt mass concentration, with 0.3 degree resolution at 35 levels (surface to 100 hPa). Several versions are available for this model, but three are used in this research: (a) the operational run (OPS) supported by U.S. Navy Fleet Numerical Meteorological and Oceanographic Command, which is used for real-time naval applications of visibility and electromagnetic propagation and features two-dimensional (2D) assimilation of NASA's

Moderate Resolution Imaging Spectroradiometer (MODIS); (b) a 2D/3D assimilation version, which combines the MODIS assimilated AOD analysis with a Fernald (1984) based extinction coefficient retrieval for CALIOP data as an assimilation constraint on the vertical model profile (Campbell et al., 2010; Zhang et al., 2010); and (c) a "free" running model, which does not apply any assimilation and is driven solely by model sources and sinks.

**2.3 Surface Reflectance/Albedo ($R$)**

Direct aerosol radiative effects are reliant on $R$ (surface albedo) of the underlying surface. The accuracy of radiative transfer modeling strongly depends on the albedo of various surfaces from ocean, land to sea ice (Lyapustin et al., 2011; Bauer and Menon, 2012; Xu et al., 2017). Therefore, albedo is also an important component of surface boundary condition in a global





weather/climate prediction system. We take an opportunity to evaluate the performance of *R* obtained from different

retrievals within the context of our radiative transfer calculations. Three main datasets are used, which are introduced below.

### 2.3.1 MAIAC

The Multi-Angle Implementation of Atmospheric Correction for MODIS (MAIAC) is an aerosol retrieval algorithm and atmospheric correction of MODIS data over land. The algorithm works globally over all surface types, although aerosols are

not currently retrieved over snow. MAIAC products include a cloud mask, water vapor, AODs and Angstrom parameters, surface spectral bidirectional reflectance factor (BRF), instantaneous BRF (iBRF), which is a specific reflectance for a given observation geometry, and albedo for MODIS land bands 1-7, and ocean bands 8-14L. The BRF and albedo are derived from the time series of 8-day measurements, and is generated uniformly at 500m and 1 km resolution in gridded format. For the purposes of this study, we utilized the BRF at 555 nm (MODIS land band 4, Lyapustin et al., 2011).


### 2.3.2 AirMSPI

The Airborne Multi-angle SpectroPolarimeter Imager (AirMSPI, Diner et al., 2007) is an eight-band (355, 380, 445, 470, 555, 660, 865, 935 nm) push-broom camera, measuring polarization in the 470, 660, and 865 nm bands, mounted on a gimbal to acquire multiangular observations over a ±67° along-track range. AirMSPI employs a photoelastic modulator-

based polarimetric imaging technique to enable accurate measurements of the degree and angle of linear polarization in addition to radiance (Diner et al., 2013). The recently developed aerosol retrieval algorithm (Xu et al., 2017) was applied to selected SEAC4RS set of AirMSPI observations. Contrary to the HSRL and the radiometers (introduced below) during SEAC[4]RS, AirMSPI was flown aboard NASA's ER-2 high altitude aircraft. This feature limited our ability to conduct the model evaluation experiment to the relatively few cases where both the DC-8 and ER-2 flew in reasonably collocated

formation.

### 2.3.3 NAVGEM and Albedo

The NAVy Global Environmental Model (NAVGEM; Hogan et al., 2014), is the U.S. Navy's operational weather model, combines a semi-Lagrangian/semi-implicit dynamical core together with advanced parameterizations of subgrid-scale moist

processes, convection, ozone, and radiation. It consists of 61 vertical levels, and a horizontal resolution of 35 km. NAVGEM meteorological fields of pressure, relative humidity, temperature and ozone are used on this research. Additionally, surface information such as surface pressure, temperature and albedo are used as input in the radiative transfer model. The albedo values used here are based on the climate data of albedo for 24 types of vegetation and bare-soil albedo at three different wavelengths that includes a factor for ground wetness change. Ocean and lake are specified with a climate

constant 0.09, and sea ice is 0.60, as well as snow being set at 0.84. The land surface climate data comes from USGS. The vegetation includes seasonal changes while ground wetness changes every time step. Nonetheless, the reflectance values contained herein are based on global distributions of seasonal variation, and have no dependence on solar zenith angle.





### 2.4 Broadband radiometers:

A pair of Kipp & Zonen CM-22 pyranometers (solar) and a pair of CG-4 pyrgeometers, (IR Broadband, BBR) were mounted on the top and bottom of the NASA DC-8 aircraft. The solar and IR BBR data are modified and calibrated as described by Bucholtz et al. (2010). The radiometers provide flight-level downwelling and upwelling irradiances between 4.5-42 nm for the BBR and 0.2-3.6 nm for SW. These irradiances are compulsory for comparing with the RTM outputs for atmospheric closure purposes. As such, our proposed experiment requires cloud-free skies in order to reconcile values between the two.

Otherwise, the in situ measurements would be contaminated to a degree that the RTM could not resolve. Despite the breadth of data collected during SEAC4RS, as such, we were limited to a single day of measurements for conducting this study.

### 2.5 Fu Liou Gu (FLG) Radiative Transfer Model

The FLG radiative transfer (RT) model is used in this study to calculate aerosol and molecular heating rates and surface/top-

of-the-atmosphere (TOA) irradiances. The FLG RT scheme (Gu et al., 2011) is a modified and improved version of the Fu-Liou RT model (Fu and Liou, 1992, 1993), which provides new and better parameterizations for aerosol properties to accommodate a more realistic radiative effects compared with observations. We utilize the delta-four-stream approximation for solar flux calculations (Liou et al., 1988) and delta-two-and-four-stream approximation for IR flux calculations (Fu et al., 1997) which are implemented in the model. The solar and IR spectra are divided into 6 and 12 bands, respectively, according

to the location of absorption bands. In addition to the principal absorbing gases listed, the calculations take into consideration absorption by the H2O continuum as well as a number of minor absorbers in the solar spectrum. Besides the aerosol, FLG requires input of atmospheric background fields (P, T, q, and $O_3$) that are obtained from NAVGEM's previous analysis time.

### 2.6 Experimental Design

HSRL profiles were matched to NAAPS in time and space. All versions of NAAPS used on this paper produced extinction (α) and AOD profiles from the surface to 100 hPa at 22 (now 35) sigma levels of variable vertical resolution (higher resolution in the lower atmosphere). In order to perform comparisons between model and observed fields, the HSRL data are "reduced" to the same model vertical resolution by employing a nearest neighbour classification constrained to model top and bottom. RT is performed using NAVGEM meteorological profiles from surface to TOA (0.1 hPa), and we assume there

is no significant aerosol loading above the 100 hPa level (aerosol layers above 100 hPa are padded to 0). This is consistent with the current HSRL observations from SEAC[4]RS, which are simultaneously constrained to aircraft height and surface elevation (the top of the HSRL observations is generally obtained within 7-10 km AGL).

Although, as will be discussed, smoke and urban aerosols are the two dominant species during this cases study, we

performed the RT calculations with each of the four particulate aerosols included in NAAPS. At this point it is important to emphasize that the NAAPS and HSRL aerosol type classifications do not necessarily overlap, so we integrate the HSRL



aerosols into the four categories that best match the NAAPS speciation: marine (sea salt for NAAPS), urban, smoke (which combines fresh smoke and smoke) and dust (dust, dusty mix, pure dust), while omitting ice (which is not an aerosol, but considered within the HSRL species). This interpretation is admittedly speculative.


Similarly, assumptions are made to match these merged HSRL/NAAPS species to the FLG optical properties tables, which are based upon the Optical Properties of Aerosol and Clouds (OPAC) software package (Hess et al., 1998) and its 18 different aerosol models. For the purposes of this study, we utilize soot (SOOT), which is used to represent non-hygroscopic absorbing black carbon of 1 g cm$^{-3}$, neglects the chain-like characteristics, and assumes no coagulation of soluble aerosol

(Hess et al., 1998). Urban aerosols represent strong pollution in continental/large city areas, for both water soluble and insoluble substances. For dust, we used mineral transported (MITR), which is used to describe desert dust that is transported over long distances with a reduced amount of large particles that are assumed not to enlarge with increasing relative humidity. Given that our observations do not coincide with a significant amount of dust, the choice of this model is considered incidental, as opposed to the compulsory effort necessary to render one or each of the other three OPAC dust

models in some combined form suitable for what is a low-order influence on the results. Finally, despite being on land, we naturally still utilize sea salt to relate what incidental amounts of marine aerosol still resolved by NAAPS.

## 3 Results

### 3.1 Description of the smoke event

Although the SEAC[4]RS field study spanned over several weeks, the necessary collocation of the aircraft observations, combined with requisite clear skies from which to most accurately apply the broadband radiometer measurements, occurred on 19 August 2013. Figure 1 shows the composite of HSRL-2 vertical profiles of aerosol backscatter coefficient at 532 nm sampled during the flight that day. The enhanced area of laser backscattering near 40° N corresponds with a transported smoke plume that serves as focus of the study.


The composite flight track in Fig. 1 depicts the HSRL taking off from Ellington Field outside of Houston, Texas (29.61° N, 95.16° W, 9.7 m MSL), through the state of Texas and the Thunder Basin Grassland in Wyoming, whose landscape contains intermingled mixed and short-grass prairies in a semi-arid climate. This flight sampled the most extensive and thick smoke plume observed during SEAC[4]RS. Within this plume, a profile with an observed peak AOD of 0.73 was sampled at 44.24°, -

104.61°, at an aircraft cruising altitude of 9.6 Km. The plume containing this profile was partially a product of large-scale smoke transport from fire activity in Wyoming, Nebraska, and South Dakota. Back-trajectory analysis for this case (not shown), demonstrate the air mass originated near the fire regions, less than 24-hrs before the research flight.

Figure 2 features the aerosol vertical distribution for the 19 August case study and its corresponding speciation. The profiles

depict a five-minute averaged HSRL segment of 532 nm aerosol extinction coefficient (km$^{-1}$) centered on 44.24° N, 104.61°



W, along with the three corresponding NAAPS 550 nm model profiles from the nearest analysis time. There are significant discrepancies in terms of the aerosol profile structure and composition comparing them. The HSRL resolves smoke mostly in the free troposphere, whereas the models constrain the layers to near the surface. Further, AODs between them differ significantly (see Table 1), with the HSRL nearly doubling each of the NAAPS runs.


The HSRL retrieval is dominated by smoke (0.30 AOD) and urban aerosols (0.42 AOD). NAAPS includes 4 types (smoke, urban, dust and sea salt as depicted on Table 1). Smoke AODs are within 10-20% of the observations depending on the run. For the speciated AOD, the largest discrepancy between model and observations is observed with the urban aerosol range, which are significantly misrepresented in all NAAPS runs (-95% in the OPS Run, -86% in the 3D Run and -67% in the

FREE run). As suggested above, dust and maritime aerosols (sea salt) are negligible in the HSRL retrievals presented on this publication, and account for only 5% of the total aerosols in the model runs.

The HSRL vs. NAAPS differences are more notable considering the vertical distribution. For all NAAPS runs, the bulk of the aerosol is constrained within 880 and 500 hPa (Fig. 2), while the HSRL discretizes two major aerosol plumes between

400 – 650 hPa associated with urban and smoke aerosols. All model versions failed at properly characterizing the aerosol heights and depth of the smoke layer (situated between 400-500 hPa). Most notably, the model fails to recognize the presence of a high and elevated urban aerosol layer. As a consequence, besides estimating the radiative budget for this case, we wanted to understand how the differences in vertical distribution between the model and the observations will translate in differences within the radiative forcing field.


**3.2 Forcing Calculations ($\nabla F$):**

RT simulations were performed after taking into consideration the solar zenith angle (SZA) at the corresponding local time and location, which are depicted in Table 1. Figures 3-6 complement Table 1, showing the results of the instantaneous

aerosol radiative forcing for the event. All four figures use the common NAVGEM atmospheric profile as input, with climatological ozone profiles but different surface $R$ values used. Figure 3 considers AirMSPI 555 nm, Fig. 4 is the AirMSPI broadband value for all 7 SW channels resolved by FLG. Fig. 5 uses the MAIAC 555 nm value and Fig. 6, uses the NAVGEM climatological albedo corresponding with the closest analysis time to the DC-8 flight. $R$ used in the four simulations are noted in Table 1.


Output irradiances are contained within the four panels of each figure: (a) downwelling shortwave radiation ($SW_\downarrow$), (b) upwelling shortwave radiation ($SW^\uparrow$), (c) downwelling infrared radiation ($IR_\downarrow$), and (d) upwelling infrared radiation ($IR^\uparrow$). Each line corresponds to a different aerosol input (green = NAAPS 3D, red =NAAPS OPS, yellow = NAAPS FREE, blue = HSRL-2). The control run (NOAER), which uses NAVGEM p, T, q, and $R$, but no aerosol, is shown in magenta. The black



circle near 300 hPa represents the corresponding observations obtained at flight level from the airborne broadband radiometers.

The surface reflectance term is of little relevance for the IR calculations. Therefore, there is no associated change in the corresponding IR irradiances across the RT retrievals (Figs. 3-6). Furthermore, because of the relatively small (or no)

concentration of larger aerosols (e.g. dust and sea salt, which are active in the IR bands), IR irradiances are primarily driven by the atmospheric state and NAVGEM's moisture and temperature profiles that initialize FLG. Notably for all cases, IR closure between 1-5% was achieved between NAVGEM meteorology and the corresponding simulated radiances compared with the aircraft measurements. The relatively minor differences in IR forcing are also quantified in Table 2, which summarizes the instantaneous radiative forcing at both the surface and TOA calculated relative to the control run.


$R$ strongly influences the SW RT estimates. From Figs. 3-6, despite obtaining near-closure in the $SW_\downarrow$ term (Figs. 3c, 4c, 5c, 6c), only the outputs with the MAIAC 555 nm BRF (Fig. 5c) approach closure in the $SW^\uparrow$. That is, here we compare radiances with the airborne NRL radiometers mounted on the DC-8. When compared to the radiometers, the HSRL $SW^\uparrow$ forcing is within 2% of the airborne radiometer measurements at flight level using the MAIAC $R$. Even with differences in

vertical aerosol distribution, the NAAPS model irradiances at flight level are within 10% of the radiometers applying the MAIAC reflectances. The values of $R$ are undoubtedly far too absorbing in the other simulations when compared to the reference and the radiometer data. Given these results, we focus on the MAIAC calculated radiative forcing for the remainder of the discussion.

Table 2 summarizes the SW aerosol radiative forcing at both the surface and TOA. The reduction in net SW radiation at the surface resulting from smoke/urban aerosols retrieved from the HSRL-2 is -33.00 W m$^{-2}$ when contrasted against the control run (to the control run (no aerosols or clouds). As expected, modelled net SW radiation at the surface are smaller due to less aerosol loading (NAAPS OPS = -22.75 W m$^{-2}$, NAAPS 3D= -10.25 W m$^{-2}$, NAAPS FREE = -4.00 W m$^{-2}$) . Magnitude wise, these results compare favourably with Stone et al. (2011), who found a surface direct SW radiative forcing between -65

to -194 W m$^{-2}$ per unit aerosol optical depth during a fire event. On the other hand, Toll et al. (2014) found a net impact greater than -100 W m$^{-2}$. Both of these studies examined forcing from very intense fires over more active source regions, however, with AOD values much higher than the current study (on the order of 1 -4 AOD). At TOA, differences in NOAER and AER irradiances are essentially constrained to the $SW^\uparrow$, inducing an overall increase in total irradiances that ranges from +240 (NAAPS Free) to +256 (HSRL) W m$^{-2}$.


Besides understanding the aerosol impact on surface and TOA irradiances, it is important to understand how differences in vertical loading (as depicted in Fig. 2) would impact the vertical distribution of irradiances at the different tropospheric levels. For example, below 900hPa NAAPS and HSRL radiances only differ by 8 to 22% in the $SW_\downarrow$ and by 10% in the





$SW^{\uparrow}$; however, it is noteworthy to mention aerosol loading is negligible at those levels in the HSRL retrieval (Fig. 2).
Moving upward in the atmospheric column, departures between HSRL and NAAPS irradiances become extremely significant; most notably in the middle troposphere (Fig. 5), and particularly in $SW_{\downarrow}$. Figure 5a depicts clearly these differences, with departures between HSRL and model generated irradiances of up to 72%. These differences compare well with the mid-tropospheric smoke/urban aerosol layer in the HSRL profile (Fig 2). Table 3 summarizes the irradiances ( $SW^{\uparrow}$, $SW_{\downarrow}$ and $SW_{\downarrow TOT}$ ) for this elevated aerosol layer between 500 – 700 hPa for the calculations with the NAAPS
profiles, HSRL and NOAER.

### 3.3 Heating Rates

Figure 7 depicts the net (total) heating rates using MAIAC 555 BRF, while Fig. 8 shows the relative differences from the NOAER run. Since these observations are not averaged in time (in other words, they are the results of a single observation),
they are better referred as "instantaneous" heating rates (IHR). Consistent with the irradiance profiles, IHR profiles similarly correlate with the distributions of each aerosol profile. NAAPS profiles show an increase in net (total) heating with respect to the control run throughout the atmospheric column, though more pronounced from 900 to 600 hPa. Heating peaks around 7 K day$^{-1}$ in the lower part of the troposphere.

The HSRL case shows a slight cooling (-0.01 to -0.09 K from 828 to 767 hPa) just below the aerosol layer and a dramatic increase in IHR associated with the aerosol layer found between roughly 700 and 200 hPa (Fig. 8). For this layer, the net heating rates exceed 18 K day$^{-1}$. Recall from Fig. 2, most of the urban/smoke aerosols detected in the HSRL algorithm are located within this layer, which corresponds to over 90% of the HSRL column AOD and relatively high absorptivity at this height. Additionally, the observations were obtained almost near the peak of solar noon (10:37 local time, cosine of the SZA
= 0.82) during boreal summer. SW IHR is mostly positive at all levels corresponding with detected aerosol. This effect is more noticeable in the HSRL profile due to higher concentration of soot and urban aerosols.

In contrast to SW, IR heating rates are relatively small and negative (i.e., cooling). As identified above, the HSRL profile does not contain dust or maritime aerosols for this case, which are otherwise highly active in the IR, so we notice a slight
warming relative to the control run. Background quantities of sea salt and dust are part of the model runs, and they are significant enough to trigger a slight warming at the surface and a slight cooling within the bulk of the aerosol layer due to emission of LW radiation. However, this cooling is only of the order of about 0.1 K. There is another area of cooling near the HSRL peak, and warming of 0.13 K near TOA.

### 3.4 Additional considerations
The RTM results described here are dependent on total aerosol loading (i.e., AOD), α, R and SZA, for which again due to the limitations of the aircraft experiment we had little clear-sky data to choose from and thus retain the BBR instruments for



evaluating column closure. However, they also exhibit a strong dependence on our choice in aerosol optical properties, to include the single scattering albedo ($\omega_o$) and particle radius. The magnitude of the aerosol forcing is highly sensitive to

absorption in the particle size range of anthropogenic aerosols (Nemesure and Schwartz, 1995), which influences these results. Recall that the entire aerosol loading within the HSRL is made up by smoke and urban aerosols, which are concentrated in the same layer. Not only are soot aerosols highly absorbing due to the presence of black carbon, prescribed by the OPAC climatology (i.e., $\omega_o$ of 0.880 at 555 nm , Hess et al., 1998), but OPAC urban aerosols also contain a significant mass density of soot (7.8 mg m$^{-3}$) and high $\omega_o$ (0.817 at 555 nm) as well.


In this study, we do not evaluate sensitivities for any of the optical properties within OPAC via FLG. In reality, this assumption is not necessarily correct, mostly because our speciation does not necessarily match OPAC, and because the HSRL speciation is re-categorized to be similar to NAAPS, as explained in Sec. 2. Therefore, errors in the obtained magnitudes might be associated with this assumption. Additionally, we recognize that the direct radiative effect of absorbing

aerosols (smoke/urban) will be different for other cases due to seasonal cycles, time of the day, aerosol loading and surface characterization. We also recognize that this is an instantaneous result within a portion of a plume, and that the diurnally averaged radiative efficiency for a smoke event might be much lower than for just an instantaneous profile. However, the key conclusion remains in the significance the vertical representation of aerosols, particularly when calculating radiances or brightness temperatures throughout the visible and IR spectra.


The modeled aerosol profiles clearly differ from the HSRL observations, in part because the aerosol prognostic model proved unable in this event to resolve aerosol loading and vertical distributions at smaller/regional scales (the model resolution used in this study is $(1°)^2$ resolution, which is equivalent to 104 km). However, it still brings added value that allows near-closure relative to the observed dataset, something we do not obtain just using the background atmosphere, as

we can observe when contrasting the aerosol forcing and heating rates results with those of the control runs.

**4 Conclusions**

We have conceptualized the aerosol radiative impact of an inline aerosol analysis field coupled with a global meteorological forecast system by applying the Fu-Liou-Gu four-stream radiative transfer model to data resolved by an offline global

aerosol transport model and operational global numerical weather prediction model, utilizing NAAPS and NAVGEM analysis and surface albedo fields. Model simulations were compared with in situ validation data collected during the NASA 2013 SEAC$^4$RS experiment, including airborne HSRL, AirMSPI, simultaneous up/down SW and broadband IR irradiance measurements, as well as NASA  MAIAC, over Wyoming in the upper central plains of the United States on 19 August 2013. Our goal is a first-order characterization of model fidelities in depicting significant aerosol forcing features in the

event that NAAPS and NAVGEM were operated in a coupled configuration, using the in situ measurements to demonstrate potential column radiative closure as a verification reference.





The results highlight significant differences between the aerosol loading and vertical distribution between the NAAPS aerosol profiles and those obtained from the HSRL observations in this unique case study. Moreover, we demonstrate the

sensitivity that different aerosol distributions exhibit on radiative fluxes and heating rates, specifically in this case associated with solar-absorbing smoke and urban aerosols. Due to the nature of the dominant aerosols in this study, most of this impact is the SW forcing and heating. We observe a reduction of the net SW radiation with the HSRL profile of  -33.00 W m$^{-2}$ , -22.75 W m$^{-2}$  with the NAAPS operational 2D-var assimilation run, -10.25 W m$^{-2}$  with NAAPS 3D-var,  and -4.00 W m$^{-2}$ with the free-running aerosol model. We additionally tested the impact that different reflectances/albedos could have in the

forcing results, using values from AirMSPI, NAVGEM (i.e., climatology) and MAIAC. Our results demonstrate that the best characterization for this case study was the one provided by MAIAC, as it was the only BRF/albedo that allowed us to achieve closure in upward shortwave irradiance, as measured with the BBR array on board the NASA DC-8.

Instantaneous heating rates for the NAAPS model runs peaked around 7 K day$^{-1}$ in the lower part of the troposphere, while

the HSRL profiles resulted in values of up to 18 K day$^{-1}$ in the middle of the troposphere. The magnitudes and vertical placement of such peaks are directly proportional to the magnitude of the aerosol loading and distribution. Furthermore, there are limitations imposed by the model resolution. Horizontally, the model is very coarse (104 Km, global domain) when compared with a single point observation. Vertically, the model resolution is higher in the lower troposphere and coarser in the middle of the atmosphere, therefore, pointing to another possible reason why the model misses the mid-tropospheric

smoke enhancement. Additionally, we acknowledge there are other factors influencing these magnitudes, to include solar zenith angle, selected optical properties and surface characterization and that these results are subject to seasonality.

We highlight two additional closing points to this study. First, this was a relatively simple experiment, achievable within the broad data collection effort that SEAC$^4$RS represented. In order to apply the airborne radiometers as a direct closure proxy

for comparing the radiative transfer simulations, though, cloud-free skies were a necessity, which severely limited how much of the SEAC$^4$RS archive we could evaluate. The community, however, needs to recognize the value in the simplicity of this effort, either through coordinated airborne study and a Lagrangian view or combined surface radiation measurements paired with high-resolution, multi-spectral lidar measurements like an HSRL that directly constrain aerosol optical properties. The pending revolution of coupled aerosol/global meteorological models will prove a ripe motivation for the aerosol community

in developing such studies and providing the vigorous verification and sensitivity analysis embraced by operational meteorological modeling groups.

That point, however, raises another obvious need with respect to the diversity of aerosol scenes and the impact on such evaluation. That is, in this case we consider smoke and urban aerosol, which are reasonably well constrained within the

OPAC database (leaving aside for the moment the evolution of smoke in transport events, and thus how well OPAC really



captures such optical properties). Dust, however, is seemingly a far more complicated consideration. OPAC contains four different dust models, and their infrared impact is something that was not a primary consideration with smoke and urban aerosols in this case. Therefore, this study likely represents a relatively simple case, and it is thus again necessary that the community invests in closure studies aimed at aerosol diversity, and particularly dust, in order to thoroughly understand

inline performance and sensitivities.

One final question for future consideration arising from this work relates to how changes in the vertical distribution of aerosol-induced forcing and heating can potentially impact a forecast cycle, particularly if heating rates of the magnitude exhibited in this case are sustained within one or several data assimilation cycles within the global modelling system. We

emphasize this potential in the context of differences in the vertical impact for NAAPS, HSRL and scale-height aerosol. The distribution of the modeled aerosols (Fig. 2) puts most of the aerosols within 700 hPa, which is a forecast level that is mostly associated with forecast of precipitation and surface temperatures; while scale height distribution of aerosols would put most aerosols within the boundary layer (BL), something that would potentially influence near-surface dynamics and diurnal cycles in a model. On the other hand, the "aerosol true" (HSRL) peak loading is in the middle of the atmosphere (~500 hPa),

which can possibly impact 1000-500 hPa thickness (influencing temperatures and mid-level jets) and advection fields. The influence of using near real-time aerosol fields in the data assimilation and NWP fields, and their sensitivity to optical properties, should be studied further, not only for absorbing aerosols, but a full aerosol suite and not constrained to a study region, but also globally.

**6 Data Availability:**

The SEAC⁴RS HSRL data used in this study can be obtained at https://www-air.larc.nasa.gov/missions/seac4rs/index.html. The NAVGEM/NAAPS profiles and surface parameters are available through the Naval Research Laboratory upon request. The MAIAC BRF/albedo data is available upon request from Dr. Lyapustin. The AirMSPI L1 data is archived at https://eosweb.larc.nasa.gov/project/airmspi; the AirMSPI aerosol data is available upon request from the AirMSPI team.


**7 Acknowledgments**

Authors MO, JRC, AB, RAF, SPB and OK acknowledge the support the NASA Atmospheric Composition Campaign Data Analysis and Modeling (ACCDAM) program mission and **S**tudies of **E**missions and **A**tmospheric **C**omposition, **C**louds and **C**limate **C**oupling by **R**egional **S**urveys program (H. Maring). Portions of this work were performed at the Jet Propulsion

Laboratory, California Institute of Technology, under a contract with the National Aeronautics and Space Administration. We thank Dr. Feng Xu and Dr. Alexei Lyapustin for providing data used in the paper.





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

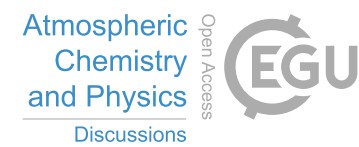

**Tables:**

| Date: | Aug 19, 2013 | | | |
|---|---|---|---|---|
| Coordinates | 44.24 ° N 104.61°W | | | |
| $\cos^{-1}(SZA)$ | 0.82 | | | |
| SFC TEMP: | 301.070 K | | | |
| SFC PRESS: | 1010.16 hPa | | | |
| MSPI 555 BRF: | 0.166678 | | | |
| MSPI BB BRF: | 0.096711 | | | |
| MAIAC 555 BRF: | 0.5152 | | | |
| NAVGEM Albedo: | 0.11000 | | | |
| AEROSOL | TOTAL AOD | SMOKE | URBAN | MARITIME | DUST |
| NAAPS | 0.40 | 0.32 | 0.06 | 0.01 | 0.01 |
| NAAPS 3D | 0.33 | 0.29 | 0.02 | 0.01 | 0.01 |
| NAAPS FREE | 0.42 | 0.25 | 0.14 | 0.01 | 0.02 |
| HSRL-2 | 0.73 | 0.30 | 0.43 | 0.00 | 0.00 |

**Table 1. Parameters utilized in the RT initialization for the Thunderbasin case study**




| | SFC | | | | TOA | | | |
|---|---|---|---|---|---|---|---|---|
| | **HSRL** | **NAAPS OPS** | **NAAPS 3D** | **NAAPS FREE** | **HSRL** | **NAAPS OPS** | **NAAPS 3D** | **NAAPS FREE** |
| **SW↑** | -215.00 | -208.25 | -200.75 | -197.00 | -259.00 | -258.25 | -248.50 | -241.00 |
| **SW↓** | -248.00 | -231.00 | -211.00 | -201.00 | 0.00 | 0.00 | 0.00 | 0.00 |
| **Total SW** | -33.00 | -22.75 | -10.25 | -4.00 | 259.00 | 258.25 | 248.50 | 241.00 |
| **IR↑** | 0.00 | 0.00 | 0.00 | 0.00 | -3.00 | -1.00 | 0.00 | -1.00 |
| **IR↓** | 0.00 | 1.00 | 0.00 | 1.00 | 0.00 | 0.00 | 0.00 | 0.00 |
| **Total IR** | 0.00 | 1.00 | 0.00 | 1.00 | -3.00 | -1.00 | 0.00 | -1.00 |
| **TOTAL (SW+IR)** | **-33.00** | **-21.75** | **-10.25** | **-3.00** | **256.00** | **257.00** | **248.50** | **240.00** |

**Table 2.** Instantaneous radiative forcing on surface and TOA for the Thunderbasin case study for MAIAC 555 reflectances







| Pressure | $\delta SW^{\uparrow}$ | | | | | $\delta SW_{\downarrow}$ | | | | | $\delta SW_{TOT}$ | | | | |
|---|---|---|---|---|---|---|---|---|---|---|---|---|---|---|---|
| | HSRL | 3D | OPS | FREE | NOAER | HSRL | 3D | OPS | FREE | NOAER | HSRL | 3D | OPS | FREE | NOAER |
| 0.504 | 215 | 201 | 193 | 214 | 461 | 904 | 1040 | 1030 | 1020 | 1060 | 688 | 839 | 837 | 806 | 599 |
| 0.572 | 236 | 200 | 194 | 215 | 461 | 809 | 1010 | 999 | 989 | 1040 | 573 | 810 | 806 | 774 | 579 |
| 0.628 | 248 | 207 | 199 | 221 | 461 | 764 | 961 | 951 | 943 | 1020 | 517 | 754 | 752 | 722 | 559 |
| 0.667 | 251 | 216 | 206 | 228 | 461 | 746 | 915 | 907 | 904 | 1010 | 495 | 699 | 701 | 675 | 549 |
| 0.708 | 252 | 226 | 215 | 237 | 461 | 737 | 868 | 863 | 866 | 997 | 485 | 642 | 648 | 628 | 536 |

**Table 3. Instantaneous SW radiative forcing in the mid-troposphere, corresponding to HSRL smoke/urban aerosol layers.**









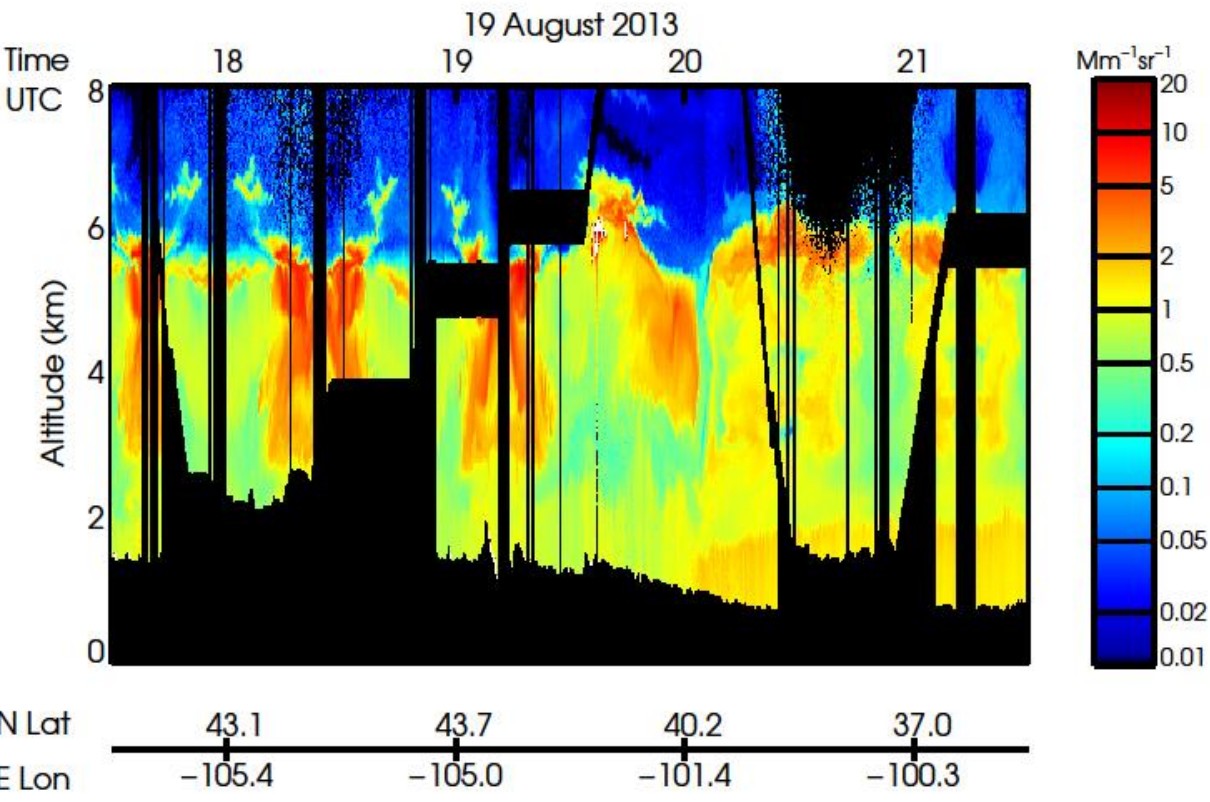


**Figure 1. Composite of DIAL/HSRL vertical profiles of aerosol backscatter coefficient at 532 µm as sampled on the research flight on 19 August 2013.**




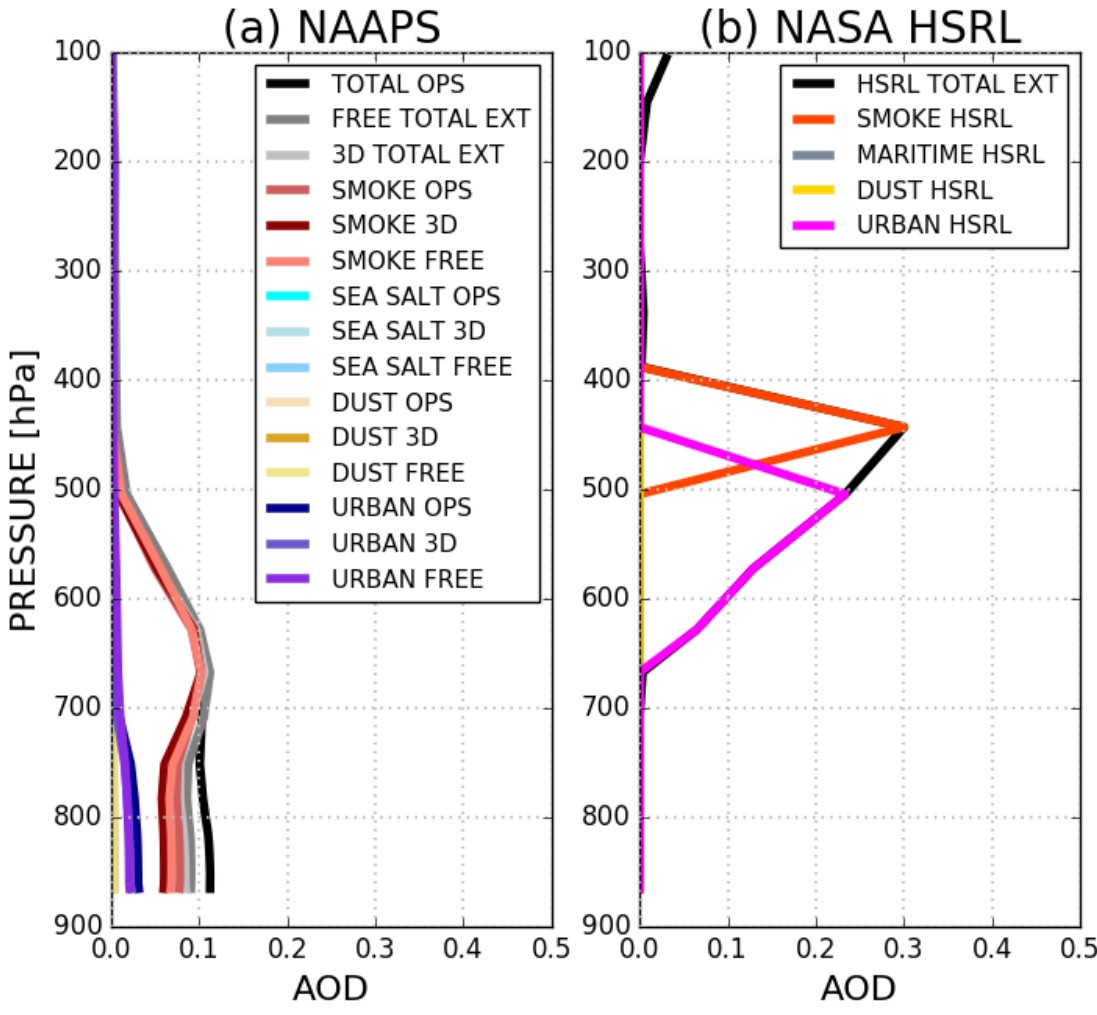

**Figure 2. (a) NAAPS and (b) HSRL aerosol vertical distribution (AODs) for the 19 August case study with corresponding aerosol speciation.**



**Figure 3: Forcing Calculations results in the (a)** $SW_\downarrow$ **(b)** $SW^\uparrow$ **(c)** $IR_\downarrow$**, (d)** $IR^\uparrow$ **using MSPI 555 nm reflectance value retrieved for the Thunder Basin Case Study, 19 Aug 2013.**



**Figure 4: Same as Fig. 3, but with MSPI BB (BRF = 0.096711).**





**Figure 5: Same as Fig. 3, but with MAIAC 555 nm (BRF = 0.5152)**





**Figure 6: Same as Fig. 3, but with NAVGEM Albedo (0.11000).**



**Figure 7. Instantaneous Heating Rates (IHR) for (a) the net, (b) SW$_{Total}$ and (c) IR using MAIAC 555 nm (BRF = 0.5152).**





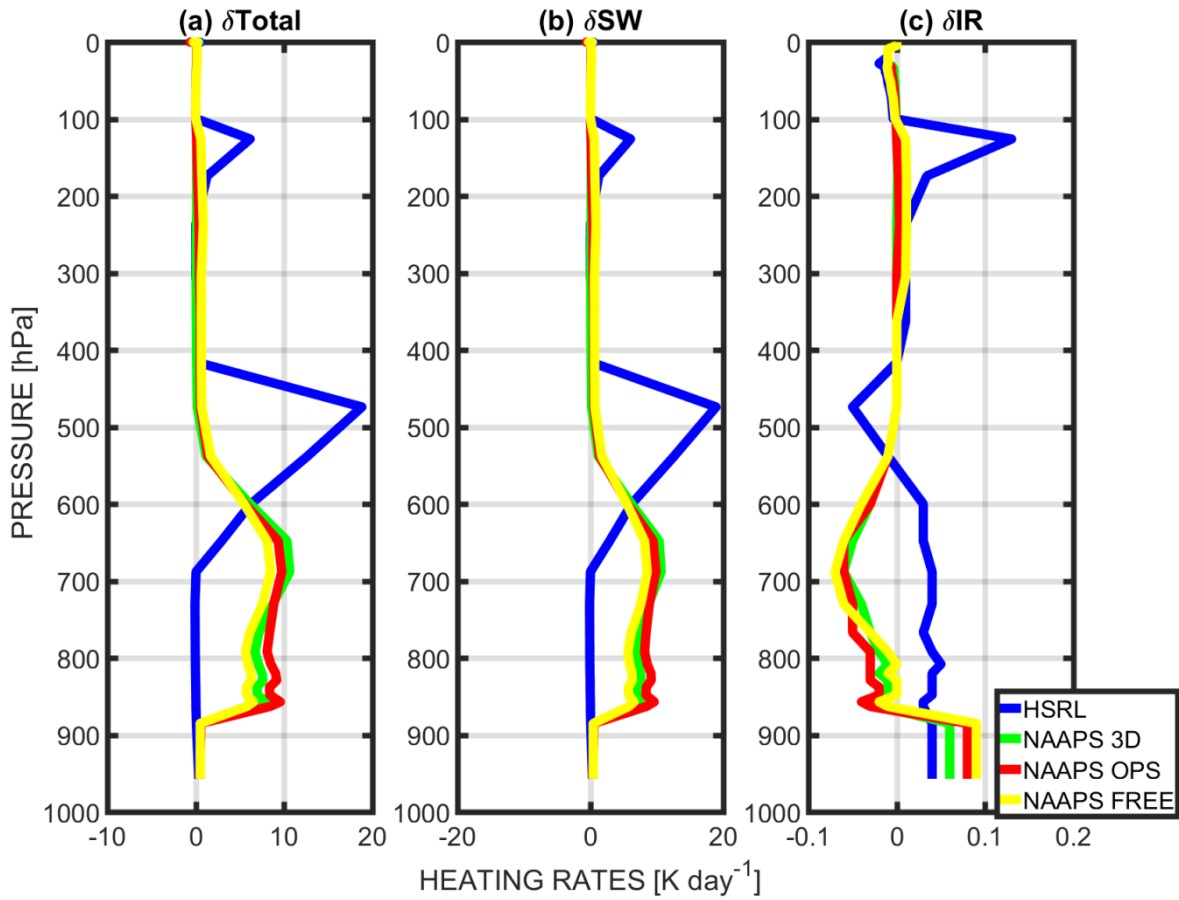

**Figure 8: Instantaneous Heating Rates Differences ($\delta(IHR) = NOAER - AER$) between all 4 aerosol profiles (AER= HSRL,**

**NAAPS OPS, NAAPS 3D, NAAPS FREE) and the control run (NOAER) using MAIAC 555 nm (BRF = 0.5152). The different**

**panels depict (a) the net instantaneous heating rates ($\delta IHR_{TOT}$), (b), the SW$_{Total}$ instantaneous heating rates ($\delta IHR_{SW}$) and (c) IR**

**instantaneous heating rates ($\delta IHR_{IR}$)**
