# Peer review of "Quantifying the Direct Radiative Effect of Absorbing Aerosols for Numerical Weather Prediction: A case study"

_Atmospheric Chemistry and Physics, 2018_

## Referee Comment (RC1) · Anonymous Referee #3 · 7 Jun 2018

This paper compares the vertical profiles and associated radiative effects of aerosols simulated by a global aerosol transport model (NAAPS) with in situ data collected during SEAC4RS. The heating rates due to aerosols are evaluated and the implications for weather prediction are discussed. The manuscript is scientifically correct and the results are well laid out. My main concern is related to the scientific significance of the paper. The following comments should be addressed prior to recommendation for publication.

Specific comments:

1. The authors highlight the potential influence of the representation of aerosols in

numerical weather prediction models thoroughout the paper. However, in the study only a comparison between modeled and observed aerosol properties is conducted while the effect on weather forecasts is not evaluated. As mentioned by the authors, the study represents a relatively simple case and the results may not be generalized to other cases. I think it would be desirable to have a more detailed discussion on the implications for weather forecasts and/or some simple experiments of the aerosol effects on numerical weather prediction models.

2. Figures 3b, 4b, 5b, 6b: Are the magenta lines correct? I would expect the lines to be different for different surface albedo. Also the values do not seem to be consistent with those in Tables 2 and 3.

3. Line 298 and Table 2: Why does the net SW radiation at the surface modeled by the three versions of NAAPS differ by more than a factor of 5, while the AOD is similar across the three models (Table 1)?

Techinical corrections:

1. Line 286: The downward SW flux is shown in Figs 3a, 4a, 5a, 6a rather than Figs 3c, 4c, 5c, 6c.

2. Tables 2 and 3: Units in these tables are missing.
* * *

---

## Referee Comment (RC2) · Anonymous Referee #1 · 7 Jun 2018

This study attempts to verify the simulated aerosol vertical profiles and the corresponding radiative responses, by using the combination of NAAPS and NAVGEM models, against the observations from the measurements of the high spectral resolution lidar (HSRL) and aircraft during the SEAR4RS field campaign. The sensitivity of aerosol radiative effects to the surface reflectance is specifically evaluated. The results in general is scientifically self-explanatory, and the scientific information in the manuscript are clearly delivered. My major concern is about the lack of sufficient discussions related to the different aerosol initializations. Below I list some issues which need to be addressed before the paper for publication.

[Figure]

Specific comments:

1. Although the authors conducted simulations with three different aerosol setups, including the OPS case, a case with 2D/3D data assimilation, and a "free" running case, there is very limited discussion about the sensitivity of aerosol vertical distribution and its radiative effect to the different aerosol setups. It is worthy performing such discussions since the differences in aerosol radiative forcing/heating rate could be notable. For example, as mentioned in lines 297-298, the net surface SW irradiances from different NAAPS versions are distinct from each other. In addition, the authors may pay attention to radiative forcing efficiency (=radiative forcing per unit AOD) under different aerosol initializations because of using different NAAPS version.

2. In light of the distinct discrepancy in aerosol vertical distribution between the HSRL and the simulations, it is expected that the associated aerosol radiative implications are greatly different between them. I suggest the authors might take a look at the radiative forcing efficiency, an index that eliminates the effect of different AOD levels. By doing such comparison between the HSRL and the simulations, the readership may get more sense about the model performance on the aerosol radiative effect.

3. Does the measured irradiance from the airborne broadband radiometers (the black dots in Figs. 3-6) match the modelling results when doing comparison between them? Put another way, what are the time scale and spatial scale of the measurements, and is it same as or close to the modeling results?

4. Throughout the whole paper the authors emphasize that the study focuses on a smoke event, but, according to Fig. 1, the realistic loadings of the smoke and urban originated aerosols are comparable at least in terms of AOD magnitude. The authors should state more carefully regarding the aerosol speciation. For example, the authors may want to rephrase the statements in lines 83-86 by taking off "smoke" related expressions.

Technical corrections:

1. Line 51: the first author name in the citation should be Mulcahy not Mulchany.

2. Line 297: miss the right parenthesis.

---

## Referee Comment (RC3) · Anonymous Referee #2 · 8 Jun 2018

This paper evaluates the aerosol model achieves column radiative closure relative to its depiction of the vertical mass concentration profile by combining NAAPS and NAVGEM models with HSRL and aircraft observation data during the SEAC4RS experiment. Overall, the results are clearly presented. However, there is little details about different aerosol forcing for the simulation and also the reasons for choosing these aerosol initializations as representatives. It is important for showing the scientific significance of the study. Also further discussion on the sensitivity of aerosol properties on the simulation results are little involved. The following comments should be addressed prior to recommendation for publication. Specific comments: 1. It is interesting for conducting simulations with different aerosol conditions (e.g. OPS, 3D, FREE), however, there is

little explanation about these experiments setups. It is necessary to let us know how your simulations are designed and why these three sensitivity tests are chosen. Also, there is little details about simulation on the case study (e.g. model configuration, initial time, output time frequency...).

2. The titles of tables and figures are too concise. The labels use abbreviation, but there is no further explanation in the titles.

3. The comparisons are all based on the observed profile with peak AOD sampled at 44.24° N, 104.61°W. How about other location and other time? Do they show the similar results?

Techinical corrections: 1. Line 286: should be Figs. 3a, 4a, 5a, 6a.

2. Table 2: no unit.

3. Table 3: no unit.

---

## Author Comment (AC1) · 16 Aug 2018

To: Reviewer 1 Atmos. Chem. Phys., http:// editor.copernicus.org/ From: Dr. Mayra I. Oyola mayra.oyola.ctr@nrlmry.navy.mil

Corresponding author for acp-2018-284

Dear Reviewer 1,

We are appreciative that you have considered reviewing our article: "Quantifying the Direct Radiative Effect of Absorbing Aerosols for Numerical Weather Prediction." for publication in the Journal of Atmospheric Chemistry and Physics. Thank you for your time

and effort to make of this a much stronger manuscript. The comments and questions (along with associated changes) have been addressed below and are also reflected in the manuscript. Reviews: 1-2 are addressed now in Section 3.4 (lines 352 - 371): "The RTM results described here are dependent on vertical distribution, total aerosol loading (i.e., AOD), $\alpha$, R and SZA, for which again due to the limitations of the aircraft experiment we had little clear-sky data to choose from and thus retain the BBR instruments for evaluating column closure. The impact of the vertical distribution has been addressed already within the context of the vertically resolved irradiances and heating rates in the previous two sections. Of significant importance is how the net surface SW radiances from different NAAPS versions are distinct from each other, even though neither column AOD, nor the aerosol vertical distributions vary dramatically between NAAPS runs. This is primarily due to differences in speciation classification among the profiles, not because of total aerosol loading. In other words, AOD is similar, but the speciation distribution is not, so this is a reflection of the radiative forcing efficiency of the aerosol. Notice on Table 1, that the distribution of urban aerosols is much higher in the NAAPS FREE than on its counterparts, constituting 33% of the total AOD. Urban aerosols only represent 15% of the total AOD in the operational run (NAAPS) and 6% in the NAAPS 3D. On the other hand, the smoke is distributed very differently (80% NAAPS, 87% NAAPS 3D, and 60% NAAPS FREE). FLG utilizes total AOD and the speciation distribution (percentage weights) in the calculations. Therefore, we believe difference in the surface (and in the net) SW radiances are strongly dependent on our choice in aerosol optical properties that are associated to the difference in speciation, to include the single scattering albedo ($\omega\_o$) and particle radius. The magnitude of the aerosol forcing is highly sensitive to absorption in the particle size range of anthropogenic aerosols (Nemesure and Schwartz, 1995), which influences these results. The same can be stated about the results with the HSRL extinction. Recall that the entire aerosol loading within the HSRL is made up by smoke and urban aerosols, which are concentrated in the same layer. Not only are soot aerosols highly absorbing due to the presence of black carbon, prescribed by the OPAC climatology (i.e., $\omega\_o$ of 0.880 at

555 $\mu$m , Hess et al., 1998), but OPAC urban aerosols also contain a significant mass density of soot (7.8 mg m-3) and high $\omega$\_o (0.817 at 555 $\mu$m) as well".

3. Another reviewer also pointed this out, within the context of better explaining the model initialization and parameters. Now it is clear on Section 2.6 that all of the NAVGEM/NAAPS profiles used as input, correspond the previous analysis time (15-18 UTC) – which means that the match between the observed radiances (aircraft) are compared to radiances calculated with profiles from the closest analysis time. The 18Z analysis is the closest to the discussed study case. I truly appreciate you asking for this clarification. The beginning of Section 2.6 now reads (lines 194-211): "HSRL aerosol observations are matched spatiotemporally to the closest NAAPS/NAVGEM analyses profiles. All versions of NAAPS used on this paper contain extinction ($\alpha$) and AOD profiles from the surface to 100 hPa at 22 (now 35) sigma levels of variable vertical resolution (higher resolution in the lower atmosphere). In order to perform comparisons between model and observed fields, the HSRL data are "reduced" to the same model vertical resolution by employing a nearest neighbour classification constrained to model top and bottom. Besides the aerosol, FLG requires input of atmospheric background fields. P, T, q, and O3 profiles are obtained from NAVGEM's previous analysis time to the flight overpass. The case study presented here (19 August 2013), uses profiles from the analyses corresponding to 15 and 18 UTC. There are four different aerosol profiles used as input: one from HSRL (taken as the true) and three that are obtained from the closest NAAPS analysis (which matches NAVGEM's analysis time). Besides extinction, both the HSRL and NAAPS datasets also contain aerosol speciation profiles. Therefore, each extinction profile is paired to a corresponding speciation profile that is matched to the FLG internal optical properties as described below. Each of the NAAPS analyses profiles correspond to a different assimilation version, as described in Section 2.2 (NAAPS 3D, NAAPS OPS, NAAPS FREE). A control run (NOAER) is set in a similar fashion, but with no aerosol feedback included. Radiative transfer calculations on FLG are performed on each profile from surface to TOA (0.1 hPa), and we assume there is no significant aerosol loading above the 100 hPa level

(aerosol layers above 100 hPa are padded to 0). This is consistent with the current HSRL observations from SEAC4RS, which are simultaneously constrained to aircraft height and surface elevation (the top of the HSRL observations is generally obtained within 7-10 km AGL)".

4. Some of this is also addressed in review statements 1-2, as discussed above. All statements pointing to smoke as a primary aerosol have been modified to include both (primarily lines 84-86 and lines 213-215). Technical corrections: Name correction in line 51 has been made to "Mulcahy et al." Line 297 has been corrected to read: "run (to the control run (no aerosols or clouds))".

Please also note the supplement to this comment:
https://www.atmos-chem-phys-discuss.net/acp-2018-284/acp-2018-284-AC1-supplement.pdf
* * *

---

## Author Comment (AC2) · 16 Aug 2018

August 15, 2018.

To: Reviewer 2 Atmos. Chem. Phys., http:// editor.copernicus.org/ From: Dr. Mayra I. Oyola mayra.oyola.ctr@nrlmry.navy.mil

Corresponding author for acp-2018-284

Dear Reviewer 2,

We are appreciative that you have considered reviewing our article: "Quantifying the Direct Radiative Effect of Absorbing Aerosols for Numerical Weather Prediction." for

publication in the Journal of Atmospheric Chemistry and Physics. Thank you for your time and effort to make of this a much stronger manuscript. The comments and questions (along with associated changes) have been addressed below and are also reflected in the manuscript. Reviews: 1. The beginning of Section 2.6 now reads (lines 194-211): "HSRL aerosol observations are matched spatiotemporally to the closest NAAPS/NAVGEM analyses profiles. All versions of NAAPS used on this paper contain extinction ($\alpha$) and AOD profiles from the surface to 100 hPa at 22 (now 35) sigma levels of variable vertical resolution (higher resolution in the lower atmosphere). In order to perform comparisons between model and observed fields, the HSRL data are "reduced" to the same model vertical resolution by employing a nearest neighbour classification constrained to model top and bottom. Besides the aerosol, FLG requires input of atmospheric background fields. P, T, q, and O3 profiles are obtained from NAVGEM's previous analysis time to the flight overpass. The case study presented here (19 August 2013), uses profiles from the analyses corresponding to 15 and 18 UTC. There are four different aerosol profiles used as input: one from HSRL (taken as the true) and three that are obtained from the closest NAAPS analysis (which matches NAVGEM's analysis time). Besides extinction, both the HSRL and NAAPS datasets also contain aerosol speciation profiles. Therefore, each extinction profile is paired to a corresponding speciation profile that is matched to the FLG internal optical properties as described below. Each of the NAAPS analyses profiles correspond to a different assimilation version, as described in Section 2.2 (NAAPS 3D, NAAPS OPS, NAAPS FREE). A control run (NOAER) is set in a similar fashion, but with no aerosol feedback included. Radiative transfer calculations on FLG are performed on each profile from surface to TOA (0.1 hPa), and we assume there is no significant aerosol loading above the 100 hPa level (aerosol layers above 100 hPa are padded to 0). This is consistent with the current HSRL observations from SEAC4RS, which are simultaneously constrained to aircraft height and surface elevation (the top of the HSRL observations is generally obtained within 7-10 km AGL)".

2. Captions have been modified to contain further explanation of what is depicted.

[Figure]

They are delineated in blue in the draft. 3. The explanation for this is given in the opening of Section 3.1, with along with the added line: "Although the SEAC4RS field study spanned over several weeks, the necessary collocation of the aircraft observations, combined with requisite of cloud free conditions from which to most accurately apply the broadband radiometer measurements, occurred on 19 August 2013. The comparisons shown in this paper are all based on this date/time, given that this was the one window of opportunity where all of the instruments were synergistically and strategically operating. Additionally, the case matched an high-loading aerosol event that warranted attention. Figure 1 shows the composite of HSRL-2 vertical profiles of aerosol backscatter coefficient at 532 $\mu$m sampled during the flight that day. The enhanced area of laser backscattering near 40° N corresponds with a transported smoke plume that serves as focus of the study. The composite flight track in Fig. 1 depicts the HSRL taking off from Ellington Field outside of Houston, Texas (29.61° N, 95.16° W, 9.7 m MSL), through the state of Texas and the Thunder Basin Grassland in Wyoming, whose landscape contains intermingled mixed and short-grass prairies in a semi-arid climate. This flight sampled the most extensive and thick smoke plume observed during SEAC4RS. Within this plume, a profile with an observed peak AOD of 0.73 was sampled at 44.24°, -104.61°, at an aircraft cruising altitude of 9.6 Km. The plume containing this profile was partially a product of large-scale smoke transport from fire activity in Wyoming, Nebraska, and South Dakota. Back-trajectory analysis for this case (not shown), demonstrate the air mass originated near the fire regions, less than 24-hrs before the research flight". Technical corrections: 1. Sentence 286 has been changed to read: "R strongly influences the SW RT estimates. From Figs. 3-6, despite obtaining near-closure in the SWâĘŞ term (Figs. 3a, 4a, 5a, 6a), only the outputs with the MAIAC 555 $\mu$m BRF (Fig. 5a) approach closure in the SWâĘŚ. That is, here we compare radiances with the airborne NRL radiometers mounted on the DC-8". 2. Units have been added to Tables 2 and 3 as requested.

[Figure]

2018.

---

## Author Comment (AC3) · 16 Aug 2018

August 15, 2018.

To: Reviewer 3 Atmos. Chem. Phys., http:// editor.copernicus.org/ From: Dr. Mayra I. Oyola mayra.oyola.ctr@nrlmry.navy.mil

Corresponding author for acp-2018-284

Dear Reviewer 3,

We are appreciative that you have considered reviewing our article: "Quantifying the Direct Radiative Effect of Absorbing Aerosols for Numerical Weather Prediction." for

publication in the Journal of Atmospheric Chemistry and Physics. Thank you for your time and effort to make of this a much stronger manuscript. The comments and questions (along with associated changes) have been addressed below and are also reflected in the manuscript. Reviews: This is discussed on (now) Section 3.5 "On the impact on NWP" "One final question for future consideration arising from this work relates to how changes in the vertical distribution of aerosol-induced forcing and heating can potentially impact a forecast cycle, particularly if heating rates of the magnitude exhibited in this case are sustained within one or several data assimilation cycles within the global modelling system. We emphasize this potential in the context of differences in the vertical impact for NAAPS, HSRL and scale-height aerosol. The distribution of the modeled aerosols (Fig. 2) puts most of the aerosols within 700 hPa, which is a forecast level that is mostly associated with forecast of precipitation and surface temperatures; while scale height distribution of aerosols would put most aerosols within the boundary layer (BL), something that would potentially influence near-surface dynamics and diurnal cycles in a model. On the other hand, the "aerosol true" (HSRL) peak loading is in the middle of the atmosphere ($\sim$500 hPa), which can possibly impact 1000-500 hPa thickness (influencing temperatures and mid-level jets) and advection fields. The influence of using near real-time aerosol fields in the data assimilation and NWP fields, and their sensitivity to optical properties, is being studied further, not only for absorbing aerosols, but a full aerosol suite and not constrained to a study region, but globally. Two follow-up publication specifically address these issue, but within the context of dust and seasalt profiles. Using a 1D-Var, biases of up to 2K in temperature and 8K in dew point were found as a function of optical depth. Additionally, the newly retrieved profiles were substantially improved when compared to aerosol observations. We are also finalizing the inclusion of aerosols perturbed satellite radiances in the Navy's data assimilation system, where we have observed significant impacts on the relative humidity and temperature innovations, and an increase of more than 20% in the number of observations that pass quality and control for all hyperspectral sensors across the board". Yes, the magenta lines are correct. The magenta lines

represent the standard albedo used operationally in every case, something that is not well-explained in the paper. Therefore it has been clarified in lines 285-287: "It is noteworthy to mention the control run does not vary the albedo and it is representative of the operational parameters used in NAVGEM (Table 1)". That is an excellent question. This has been addressed now in lines 353-371: "The RTM results described here are dependent on vertical distribution, total aerosol loading (i.e., AOD), $\alpha$, R and SZA, for which again due to the limitations of the aircraft experiment we had little clear-sky data to choose from and thus retain the BBR instruments for evaluating column closure. The impact of the vertical distribution has been addressed already within the context of the vertically resolved irradiances, and heating rates in the previous two sections. Of significant importance is how the net surface SW radiances from different NAAPS versions are distinct from each other, even though neither column AOD, nor the aerosol vertical distribution vary dramatically between NAAPS runs. This is primarily due to differences in speciation classification among the profiles, not because of total aerosol loading. In other words, AOD is similar, but the speciation distribution is not. Notice on Table 1, that the distribution of urban aerosols is much higher in the NAAPS FREE than on its counterparts, constituting 33% of the total AOD. Urban aerosols only represent 15% of the total AOD in the operational run (NAAPS) and 6% in the NAAPS 3D. On the other hand, the smoke is distributed very differently (80% NAAPS, 87% NAAPS 3D, and 60% NAAPS FREE). FLG utilizes total AOD and the speciation distribution (percentage weights) in the calculations. Therefore, we believe difference in the surface (and in the net) SW radiances are strongly dependent on our choice in aerosol optical properties that are associated to the difference in speciation, to include the single scattering albedo ($\omega\_o$) and particle radius. The magnitude of the aerosol forcing is highly sensitive to absorption in the particle size range of anthropogenic aerosols (Nemesure and Schwartz, 1995), which influences these results. The same can be stated about the results with the HSRL extinction. Recall that the entire aerosol loading within the HSRL is made up by smoke and urban aerosols, which are concentrated in the same layer. Not only are soot aerosols highly absorbing due to the presence of black carbon,

prescribed by the OPAC climatology (i.e., $\omega$_o of 0.880 at 555 $\mu$m , Hess et al., 1998), but OPAC urban aerosols also contain a significant mass density of soot (7.8 mg m-3) and high $\omega$_o (0.817 at 555 $\mu$m) as well". Technical corrections: Sentence 286 has been changed to read: "R strongly influences the SW RT estimates. From Figs. 3-6, despite obtaining near-closure in the SWâĘŞ term (Figs. 3a, 4a, 5a, 6a), only the outputs with the MAIAC 555 $\mu$m BRF (Fig. 5a) approach closure in the SWâĘŚ. That is, here we compare radiances with the airborne NRL radiometers mounted on the DC-8". Units have been added to Tables 2 and 3 as requested.
* * *

---

## Author Response (AR2)

November 21, 2018.

To: Yun Qian, (yun.qian@pnnl.gov )
Editor, Atmos. Chem. Phys., http:// editor.copernicus.org/
From: Dr. Mayra I. Oyola
Corresponding author for acp-2018-284

Dear Yun,

We are appreciative that you have considered reviewing our article: "Quantifying the Direct Radiative Effect of Absorbing Aerosols for Numerical Weather Prediction." for publication in the Journal of Atmospheric Chemistry and Physics. We would also like to thank the reviewers for their time and effort to make of this a much stronger manuscript. The technical changes required have been addressed below and are also reflected in the manuscript.

Please do not hesitate to contact us with further questions.

Respectfully,

Mayra I. Oyola, PhD
Naval Research Laboratory
7 Grace Hopper Ave, MS2
Monterey, CA, 93940.
mayra.oyola.ctr@nrlmry.navy.mil

Technical corrections:
1. Section 3.4: Some recent work (e.g. Allen et al. 2012; Shen and Ming 2018) on the effect of aerosol absorption on the large-scale circulation, as well as its height dependence, might be helpful.

Thanks for suggesting these excellent reads. Not only they make the paper stronger, they were delightful to read.

At the beginning of Section 3.5, now reads: "Aerosol impacts large-scale circulation by virtue of its absorption and vertical distribution. For example, Allen et al. (2012) suggested that the tropical belt expansion may not be driven not only by stratospheric cooling, but also by mid-latitude heating sources due aerosol distribution. Additionally, Shen and Ming (2018), examined how aerosol absorption affects the extratropical circulation by analyzing the response to a globally uniform increase in black carbon, and suggested absorbing aerosols are capable of altering synoptic-scale weather patterns. These studies, among many others, show these impacts are dependent on the aerosol height, stressing the necessity of better constraining model-simulated aerosol vertical distributions with satellite and field measurements"

2. Figure 2: Why does the figure start from 900 hPa? I wonder if the aerosol concentration is actually very small below 900 hPa (as suggested in Section 3.2) or if there is no data?

No data below that level due to elevation (that is the surface level in that region).

3. Figure 3: It might be helpful to add "(BRF = 0.166678)" in the caption.

BRF = 0.166678 has been included as part of the caption.

References:
Allen, R. J., S. C. Sherwood, J. R. Norris, and C. S. Zender (2012): The equilibrium response to idealized thermal forcings in a comprehensive GCM: implications for recent tropical expansion. Atmospheric Chemistry and Physics, 12 (10), 4795–4816, doi:10.5194/acp-12-4795-2012.
Shen, Z., and Y. Ming (2018), The Influence of Aerosol Absorption on the Extratropical Circulation, Journal of Climate, 31 (15), 5961–5975, doi:10.1175/JCLI-D-17-0839.1.

References have been added. Thank you very much!!!